# Endotoxin in Sepsis: Methods for LPS Detection and the Use of Omics Techniques

**DOI:** 10.3390/diagnostics13010079

**Published:** 2022-12-27

**Authors:** Grazia Maria Virzì, Maria Mattiotti, Massimo de Cal, Claudio Ronco, Monica Zanella, Silvia De Rosa

**Affiliations:** 1Department of Nephrology, Dialysis and Transplant, San Bortolo Hospital, 36100 Vicenza, Italy; 2IRRIV—International Renal Research Institute Vicenza, 36100 Vicenza, Italy; 3Nephrology, Dialysis and Renal Transplant Unit, IRCCS—Azienda Ospedaliero-Universitaria di Bologna, Department of Experimental Diagnostic and Specialty Medicine (DIMES), Alma Mater Studiorum University of Bologna, 40126 Bologna, Italy; 4Centre for Medical Sciences—CISMed, University of Trento, Via S. Maria Maddalena 1, 38122 Trento, Italy; 5Anesthesia and Intensive Care, Santa Chiara Regional Hospital, APSS Trento, 38122 Trento, Italy

**Keywords:** endotoxin, LPS, sepsis, omics, proteomics

## Abstract

Lipopolysaccharide (LPS) or endotoxin, the major cell wall component of Gram-negative bacteria, plays a pivotal role in the pathogenesis of sepsis. It is able to activate the host defense system through interaction with Toll-like receptor 4, thus triggering pro-inflammatory mechanisms. A large amount of LPS induces inappropriate activation of the immune system, triggering an exaggerated inflammatory response and consequent extensive organ injury, providing the basis of sepsis damage. In this review, we will briefly describe endotoxin’s molecular structure and its main pathogenetic action during sepsis. In addition, we will summarize the main different available methods for endotoxin detection with a special focus on the wider spectrum offered by omics technologies (genomics, transcriptomics, proteomics, and metabolomics) and promising applications of these in the identification of specific biomarkers for sepsis.

## 1. Introduction

Sepsis is a life-threatening multiple organ dysfunction, resulting from a deregulated host response to infection [1], which could progress into acute respiratory distress syndrome (ARDS), acute kidney injury (AKI), or disseminated intravascular coagulation (DIC) [2]. It is estimated that the prevalence of sepsis is 31.5 million patients per year with 5.3 million deaths per year. High-income countries’ hospital mortality rates for general and severe sepsis are significantly elevated (17% and 26%, respectively) [3]. The annual medical cost for 230,000 patients with sepsis treated in the ICU is about USD 4.6 billion, and the related medical and social load is very high [4,5,6]. Furthermore, because of an increasingly aging society in many countries, the occurrence of sepsis is likely to be on the rise. Although guidelines for the diagnosis and treatment of sepsis made great progress in the past decade and the prognosis has improved, the mortality rate is still high [7]. A deep understanding of underlying mechanisms, early and accurate diagnoses, and adequate treatments of sepsis is essential for improving sepsis management.

The pathogenesis of sepsis is highly multifaceted and it involves several different mechanisms, such as infection, inflammation, immune system activation, blood coagulation, dysfunction of endothelium, and tissue damage through cell death and/or apoptosis [8,9,10]. In the first phases, sepsis is characterized by an exaggerated systemic inflammatory immune response and cell death through apoptosis; on the contrary, in the later stages, sepsis is characterized by progressive immunosuppression, also known as immune paralysis. In this context, pro-inflammatory reactions are activated with the aim of removing invading pathogens, while anti-inflammatory responses are activated to limit local and systemic tissue injury and damage. The net sum of these antagonistic processes may result in cell death and consequent tissue dysfunction and damage until organ failure [8,9,11,12,13,14,15,16].

### 1.1. Diagnosis of Sepsis

The latest sepsis definition [1] highlights the potential lethality of sepsis and the need for urgent recognition in order to start a prompt and appropriate therapy to interrupt the underlying pathological mechanism. The clinical presentation of sepsis is heterogeneous and aspecific; currently, there are no specific clinical markers of the activation of the dysregulated host response. Fortunately, a score that is easily calculated bedside and integrated with an easy and reliable laboratory test could permit the early recognition of the state of inflammation (SIRS) [17] and organ dysfunction (SOFA score) [18,19]. It has been recently proven that the severity of organ dysfunction correlates with the prognosis of sepsis [20], but there are no specific markers for therapy stratification.

Although the etiology of sepsis is unknown in one-half of cases, the most common trigger for sepsis and septic shock are Gram-positive bacteria (*Staphylococcus aureus* and *coagulase negative Staphylococci*) followed by Gram-negatives (i.e., *Escherichia coli*, *Klebsiella pneumoniae*, *Enterobacter* spp., *Acinetobacter baumannii*, and *Pseudomonas aeruginosa*) but also other microorganisms such as mycobacteria and fungi (*Candida*, *Histoplasma*, and *Aspergillus*). Further different types of viruses and protozoa can be isolated from the blood of immunocompromised septic patients [21]. In addition, antimicrobial-resistant bacteria, such as methicillin-resistant *Staphylococcus aureus* (MRSA) and vancomycin-resistant *Enterococci*, were detected in septic patients with nosocomial acquisitions of infection.

Effective therapy is the cornerstone for early recovery and improved and more favorable prognosis; therefore, in patients with suspected sepsis or septic shock, a microbiological analysis should be performed as soon as possible. Unfortunately, standard culture-based microbiology techniques often yield results within 48–96 h, and in one-half of cases no microbiological isolation is available; therefore, the implementation of methods for rapid detection and identification may provide clinical and economic benefits enabling clinicians to choose a timely targeted therapy.

Many serological biomarkers helpful for sepsis diagnosis and as outcome predictors are routinely employed (i.e., procalcitonin, c-reactive protein, lactate serum level, and cytokines), but they lack specificity, not able to distinguish sepsis from other conditions [22]. The crucial role of endotoxin, also known as lipopolysaccharide (LPS), in Gram-negative sepsis has been already established. Increasing evidence shows that LPS mediates the early direct injury to multiple cell lines of the host by the suppression of the transcription of genes involved in ribosomal function and translation and mitochondrial processes, and by triggering inflammatory cell systems. Moreover, the endotoxin level seems to correlate with different sepsis phases in murine models [23]. Therefore, endotoxemia determination represents a useful tool for sepsis diagnosis.

### 1.2. Gram-Negative Sepsis and Endotoxin

Systemic Gram-negative sepsis is one of the most severe diseases for hospitalized patients, mainly if critically ill. Endotoxin, the major initiator of the process, is the main component of the outer membrane of Gram-negative bacteria [3], first discovered in the 1800s. There are different forms of LPS, produced by distinct and specific species of bacteria, and also LPS toxicity is different between bacterial species. A small amount of LPS may stimulate the immune system, inducing a very strong activation against infection, while a high amount of LPS in the blood may provoke harmful septic shock syndromes. For example, endotoxin may trigger cellular biosynthesis, activate intracellular mechanisms of apoptosis, and induce strong activation of inflammatory pathways with the consequent release of pro-inflammatory cytokines and chemokines, such as tumor necrosis factor-alpha (TNF-α), interleukin-6 (IL-6) and interleukin-18 (IL-18), and other bioactive metabolites of organ damage and septic shock [3,22,23].

In this review, we briefly describe the endotoxin molecular structure and its main pathogenetic action during sepsis. In addition, we summarize the main different available methods for endotoxin detection, including the wider spectrum offered by omics technologies, which include genomics, transcriptomics, proteomics, and metabolomics. The extensive data collection obtained with these technologies could be very helpful in the identification of specific biomarkers for sepsis.

Bibliographic databases are the main source for finding information for medical interventions and clinical innovations. Comprehensive research in the PubMed and Cochrane databases was performed using the following search string: (endotoxin OR LPS) AND (sepsis OR Gram-negative), (endotoxin OR LPS) AND (structure OR characteristics), (endotoxin OR LPS) AND (therapy), (endotoxin OR LPS) AND (sepsis OR Gram-negative), (endotoxin OR LPS) AND (proteomics OR omics), (endotoxin OR LPS) AND (test OR detection OR method OR methodology). Furthermore, PubMed was used to identify narrative or systematic reviews and publications using specific terms to elaborate and add details to our results. The references of the retrieved papers were used to find more literature and references.

## 2. Endotoxin: Structure and Characteristics

The external membrane of Gram-negative bacteria is characterized by an asymmetric structure: the inner cytoplasmic membrane wall is rich in phospholipids. On the contrary, the outer leaflet contains a high volume of specific LPS, accounting for 75% of the external surface of Gram-negative bacteria (Figure 1a). The space that separates the outer membrane from the inner membrane is called the periplasmic space. LPS is a macromolecular glycolipid (MW 10–20 kDa) and it is composed of three different domains that vary genetically, structurally, and antigenically: lipid A, a hydrophobic membrane anchor; the core oligosaccharide, a short chain of sugar residues with multiple phosphoryl substituents; and the O-antigen, a distal serospecific polymer composed of polysaccharide units [4,13].

Lipid A (endotoxin) is highly conserved among different species. It is a glucosamine-based phospholipid that makes up the outer monolayer of the outer membranes of most Gram-negative bacteria, and it consists of a phosphorylated N-acetyl glucosamine dimer linked with six to seven saturated fatty acid chains. In most cases, some fatty acids are directly attached to N-acetyl glucosamine dimer, but sometimes they could be esterified. The lipid A chain has a huge architectural variability between different bacterial species. Variations can be in terms of the number and length of acyl chains, or there may be other substituting groups at the positions of phosphate moieties [24]. Lipid A is the bioactive domain of LPS (Figure 1b): it can be detected at picomolar levels by an ancient receptor of the innate immune system present on macrophages and endothelial cells of animals (Toll-like receptor 4, TLR4) [25], activating signal transduction involving inflammatory pathways (TNF-α, IL1-β, tissue factor) and responsible for the activation of the inflammatory storm at the basis of LPS toxic effect and of its myriad in vivo and in vitro actions [4].

The central role of lipid A in the pathogenesis of sepsis makes it a good target for antibiotic strategies [26].

The core oligosaccharide can be divided into two regions: the inner core (lipid A proximal) and outer core. The outer core region provides an attachment site for O-polysaccharide (O-antigen). Within a genus or family, the structure of the inner core tends to be well-conserved, suggesting the importance of the core in outer membrane integrity [27]. The limited structural variation in the core oligosaccharide within a genus is in striking contrast to the hypervariable O-polysaccharides and has stimulated interest in the possibility of targeting the core oSs for the generation of immunotherapeutic antibodies. It is characterized by two different portions: a hydrophilic polysaccharidic chain responsible for its immunogenicity; and the O-antigenic, a periodically repeating hydrophilic polysaccharide unit. The O-polysaccharide repeat unit structures can differ in the monomer glycoses, the position and stereochemistry of the *O*-glycosidic linkages, and the presence or absence of noncarbohydrate substituents. O-repeat units from different structures may comprise varying numbers of monosaccharides, they may be linear or branched, and they can form homopolymers (i.e., a single monosaccharide component) or, more frequently, heteropolymers [27]. The structure of the O-polysaccharide defines the O-antigen serological specificity in an organism, even if the numbers of unique O-antigens within a species vary considerably. The primary role(s) of the O-polysaccharides appear to be protective: they can up-regulate bacterial intracellular survival, they may contribute to bacterial evasion of host immune responses, particularly the alternative complement cascade [28], they could defend the cell from oxidative stress, they could avoid the internalization inside host epithelial cells, and they can contribute to bacteria motility. The immunogenicity of the O-antigen polysaccharide evokes a great immunity response intermediated by specific antibodies [3,4,9].

LPS can be present in two different forms: “rough”, including only lipid A and core subunits, or “smooth”, including all three aforementioned units (LPS capped with O-antigen). For its chemical and structural features, LPS has very good heat stability and a worthy resistance to oxidative stress. The production site of LPS is located in the internal membrane of the bacterial cell, and it needs to be transferred from the inner to the outer membrane and to the bacterial surface (final position). This mechanism involves a specific transport pathway, mediated by a protein complex composed of seven different proteins. This protein complex is a sort of bridge helping LPS to cross the periplasmic space and reach the outer membrane. Precisely, a beta-barrel membrane protein allows the transport of LPS to the leaflet of the outer membrane. Finally, bacterial wall shedding and bacterial lysis allow the release of LPS into the host blood circulation. Endotoxin binds the host receptor Toll-like receptor 4 (TLR4), which is characterized by a big, leucine-rich extracellular domain, a single transmembrane segment, and a small cytoplasmatic tail. TLR4 is located on the surface of various cells (for example neutrophils, monocytes, and macrophages). TLR4 creates a heterodimer with co-receptor MD-2, and together they are involved in a common pattern for LPS recognition [10,13,23].

In this setting, identification, determination, quantification, and monitoring of LPS are crucial and they are performed via the detection of LPS receptors as well as other accessory proteins. Among the latter, CD14 (cluster of differentiation 14) has a prominent role: it binds LPS in the presence of soluble lipopolysaccharide-binding protein LBP, making it one of the most often used for the indirect detection of LPS.

## 3. Endotoxin Removal via Extracorporeal Therapies

Proven the central role of endotoxin in Gram-negative sepsis pathogenesis, several methods of extracorporeal removal have been recently implemented as bridge treatments until satisfactory bacterial clearance through antimicrobial therapy is achieved [29]. Table 1 summarizes the main results of available and ongoing randomized control trials (Table 1).

One of the most widely used endotoxin extracorporeal blood removal therapies is adsorption with Toraymyxin^®^ (Toray, Tokyo, Japan), a polystyrene-derived cartridge with molecules of polymyxin B (PMX-B) covalent bound. Polymyxins are cyclic cationic polypeptide antibiotics derived from *Bacillus polymyxa* with an effective antimicrobial activity against Gram-negative bacteria, but their clinical use has been limited for their nephrotoxicity and neurotoxicity [39]. Polymyxins can also bind lipid A with a very stable interaction with its hydrophobic residues and neutralize endotoxin filtered by blood flowing through the extracorporeal circuit inside the cartridge, avoiding toxic systemic effects. Furthermore, it has been recently observed that PMX-B contribute to reducing inflammatory storm through different mechanisms: entrapment of monocytes and neutrophils, and clearance of cytokines, such as TNF-α and IL-6. Although routinely used in Japan for patients with a Gram-negative bacteria infection, conflicting results are currently available about its impact on mortality [30,31,32,33,40]. Some randomized clinical trials (RCTs) comparing polymyxin B adsorption to a standard treatment suggest a beneficial effect of Toraymyxin^®^ on severe patients, patients with endotoxin activity levels (as evaluated by the endotoxin activity assay) between 0.6 and 0.9, or those presenting a particular genetic profile [29,41,42].

Another device developed for LPS removal is the Alteco^®^ LPS adsorber (Alteco Medical AB; Lund, Sweden), which contains a synthetic peptide with adsorptive properties. The peptide is linked to the surface of a porous polyethylene matrix in order to provide an optimal binding surface. Additional conflicting results are available in the literature with evidence of a few case series of hemodynamic improvement associated with its use [43]. The ASSET (Abdominal Septic Shock–Endotoxin Adsorption Treatment) multicenter RCT was terminated early because of patient recruitment issues [34].

The oXiris^®^ hemofilter (Baxter, Meyzieu, France) was recently developed to enhance the adsorptive properties of the AN69ST membrane [44]. This membrane is made up of three different layers: AN69ST (acrylonitrile and sodium methallylsulfonate molecules); PEI (polyethyleneimine), able to adsorb large negatively charged molecules, such as endotoxins; and 4500 UI/m^2^ of heparin, to reduce thrombogenic local stimulus. If compared to the previous device, the oXiris^®^ membrane shows the advantage of being used in septic patients with AKI. Encouraging results are expected from the use of the oXiris^®^ hemofilter during sepsis management, but RCTs are needed to further confirm these results. One crossover trial comparing oXiris^®^ and standard ST-150 membrane (NCT 02600312) and another randomized trial comparing oXiris^®^ with Toraymyxin for endotoxin removal (ENDoX study; NCT 01948778) have been recently completed, and the results show new insights in the use of the oXiris^®^ membrane in sepsis and septic patients. Other ongoing RCTs aim to study the role of oXiris^®^ in other clinical contexts [37,38]. The other unsolved question lies in the timing of extracorporeal treatment: oXiris^®^ is probably more effective in the early phases of sepsis, with the aim of limiting the host immune response.

## 4. Omics Techniques for Sepsis

Given the high burden of morbidity and mortality worldwide and the diagnostic challenge related to sepsis, in the last decade, a multitude of omics techniques have been developed to better understand and clarify general pathways activated during sepsis, in the general population. Some examples of these techniques are genomics [13,45], transcriptomics [46,47,48], proteomics [45], and metabolomics [49,50].

An integrated approach of multi-omic data could be helpful in discovering molecular dynamics and pathways implicated in the pathophysiology of human disease, thus leading to innovative strategies for their early detection, specific treatment, and effective prevention. However, all of these technologies and approaches show some advantages and limitations [7]. In particular, ‘omic’ technologies allow a comprehensive understanding of the molecules, cells, and tissues. Several studies have shown the advantage of the integration of multi-omics datasets applied to a wide range of biological problems, helping to unravel the underlying mechanisms at the multi-omics level [49].

The main intent is the extensive recognition through “omic” technologies of genes (genomics), mRNA (transcriptomics), proteins (proteomics), and metabolites (metabolomics) in a specific biological sample in a non-targeted and non-biased method. Given the large quantity and the vast array of records and data produced by omics studies and technologies, it is conceivable that bioinformatics and biostatistics have a central role in sorting out and grouping the broad collection of results. Precise validation and careful analyses are the cornerstones of excluding random results from these analyses [50,51,52,53].

Today, omics technologies are advancing rapidly, and many datasets can be extrapolated from both individuals and patient populations [49]. Furthermore, because of the complex nature of sepsis, omic analysis, integrated with clinical input disease characteristics, can be useful to identify pathologically relevant biomarkers. For example, omics can be generally helpful in the sepsis context for discovering (1) biomarkers to distinguish between infectious and non-infectious causes, (2) prognostic indexes, (3) biomarkers linked with therapy, and (4) markers to predict individual patient response to therapy and to apply precision medicine [54].

### 4.1. Proteomics

Since the completion of the Human Genome Project and the accumulation of extensive genomic data, proteomics have become an integral component of the post-genomic era [7]. Specifically, with the expansion of omics methodologies, the investigation has moved to the study of the translation “products” of cellular proteins and RNA transcripts. Proteomics is a new and powerful discipline aimed at the study of the whole proteome—the sum of all proteins of an organism, tissue, cell, or biofluid—or a subfraction of it under specific and precise conditions. Finally, proteomics allows the description of expressed proteins and their modulations in different situations. The principle of proteomics is to investigate the features of proteins on a large scale, analyzing different aspects, including protein identification, post-translational modification (PTMs; glycosylation, phosphorylation, etc.), and protein function determination [55]. There are many research methods for proteomics. They include chromatography-based techniques (traditional techniques), such as two-dimensional gel electrophoresis, liquid chromatography, and mass spectrometry, with high sensitivity and resolution, and protein chip techniques (advanced technologies) [56,57]. The application and the integration of this assortment of techniques permit the identification and quantification of proteins and peptides in tissues and biological fluids, offering novel perceptions of disease-related mechanisms and progression at the cellular and molecular levels [58]. In the setting of critically ill and septic patients, proteomics is applied for the search of specific biomarkers for early diagnosis of disease [59,60,61,62,63].

Biological samples applied in sepsis proteomics are very varied and can comprise body fluids (for example plasma, serum, and urine), tissues or organs (such as hepatic tissue, cardiac tissue, and muscle), cells (such as platelets, lymphocytes, monocytes, and endothelial cells), organelles (mitochondria), and exosomes. Each biological sample has its pro and con characteristics [64]. In the context of proteomic research, the role of emerging biomarkers in sepsis is a promising area of future research. There are two different typical approaches for proteomics in sepsis. The first method is the search for biomarkers with proteomic approaches, which focus on the prompt and early diagnosis of sepsis and organ function damage [63]. The second approach includes the investigation of the molecular pathways involved in sepsis pathogenesis and sepsis-related organ dysfunction injury. This line is focused on the alterations and on the dynamic changes in protein expression when comparing septic populations and control subjects to identify therapeutic targets, thereby achieving precision medicine [7]. For example, for septic patients, the early detection of single organ failure (i.e., renal function impairment before acute kidney injury) or multi-organ syndrome is fundamental to starting timely treatment and limiting the progression of organ damage.

### 4.2. Endotoxin Detection

Based on the central role of LPS in septic conditions, recently, several methods and numerous devices for its detection have been produced. At this moment, simple, speedy, extremely sensitive, and specific tests for endotoxin determination are established and commercially available. In this paragraph, we briefly summarize the most common tests for LPS detection and the underlying methods and technologies. Table 2 reports the characteristics of LPS tests.

Rabbit Pyrogen Test. The rabbit pyrogen test was the first method approved by US Food and Drug Administration. This test is based on the measurement of the increase in a rabbit’s temperature after exposition (injection in the rabbit) with a test solution with possible contamination of pyrogenic molecules [13]. Several disadvantages of this test are obviously linked to the need of rabbits for endotoxin determination. Furthermore, this determination takes a long time and many animals are necessary for the test; it is an inadequate method for the identification of pyrogens in a clinical setting. However, the most important limitation is the qualitative result offered by this method, which does not allow for the quantification of endotoxin.

LAL (Limulus Amebocyte Lysate) Test. The limulus amebocyte lysate (LAL) test is a user-friendly test and is one of the most commonly employed tests. When exposed to LPS, amoebocytes extracted from horseshoe crabs’ blood develop a clot as a result of protease cascade activation. Briefly, the reagent is combined with equal volumes of the serially diluted test sample. After incubation at 37 °C, the mixture is verified for the presence of a clotting reaction (gel clot). Tested samples can be thereafter compared to parallel dilutions of a reference LPS. The evidence of a gel clot underlines the presence of bacterial LPS in the analyzed specimen and, in this case, the LAL test is positive [14,15].

LAL tests must be accurately controlled: all testing supplies must be pyrogen-free and experimental conditions (temperature, pH, and reaction time) must be tightly controlled. Several test kits are commercially available. The LAL test is mostly applied for pyrogen control of pharmaceutical products and is predominantly used for the evaluation of Gram-negative contamination of foodstuffs, in particular for fresh meat, milk, and eggs. The LAL test is easy to use and economic, but several factors may affect the sensitivity of the assay. For example, β-(1,3)-D-glucan, typical of fungi, algae, and yeast, may interfere with the coagulation cascade compromising LAL test results [69].

Recently, novel and innovative technologies, such as chromogenic [16], turbidimetric [70], or viscometric [71] methods, have been introduced to improve the accuracy of the LAL test.

Biosensors. In recent years, many efforts have been undertaken to find reliable methods to detect the level of LPS based on the endotoxin-affinity mechanism with notable results for LPS detection. 

A biosensor is an analytical device that elicits a measurable signal proportional to the concentration of the target molecule, usually incorporating a biological sensing element and measuring signals induced by biological interactions (Figure 2). Generally, a biosensor is composed of two main components: a biological recognition element and a signal element. The first element is used to identify the target molecule. The second one translates the biological recognition into a physically measurable signal. Finally, biosensors are specific devices for the detection of the target analyte of interest and its variations [13,72]. Biosensors are easy-to-use, rapid, and highly sensitive tests, with high selectivity for specific molecules. Furthermore, the type of biomolecule used can vary widely. Commonly, biosensors are based on the biological interactions between the sensing element and the target. High selectivity for the target molecule among a matrix of other chemical or biological components is a key requirement of the bioreceptor. Biosensors include molecules with different types of interactions, between antibodies/antigens, enzymes/ligands, nucleic acids/DNA, cellular structures/cells, or biomimetic materials [73,74,75]. A central feature of biosensors’ structure is the mechanism of connection of the biological element to the sensor surface (for example metal, polymer, or glass). The simplest way is to functionalize its surface in order for it to be easily coated by the biological element. 

After the recognition of the target, the biosensor activates a signal via a transducer element, which is translated into a measurable signal. These transducers can be optical, electrochemical, or mechanical; therefore, biosensors can be classified into electrochemical biosensors, optical biosensors, electronic biosensors, piezoelectric biosensors, gravimetric biosensors, pyroelectric biosensors, and magnetic biosensors [76].

Biosensors are giving a significant contribution to the progress of next-generation medicines. In particular, in the last years, different types of biosensors with affinity to LPS (protein-based biosensors, peptide-based biosensors, and synthetic substrates) have been developed and applied for LPS detection. In particular, electrochemical or optical biosensors are more often employed.

Currently, antibody-based biosensors are revolutionary diagnostic tools in the scenario of biosensors. In fact, antibody-based biosensors offer a sensitive and rapid analytical method for the recognition of a vast array of pathogens and their associated toxins. These biosensors take advantage of high specific binding affinity between the antibody and the related specific compound or antigen (lock and key fit mechanism) [57]. For LPS detection, antibody-based biosensors are superior in terms of specificity if compared to protein-based biosensors, which can have a cross-bind/cross-reactivity to structurally similar molecules, but, unfortunately, they are more expensive and time-consuming tests [13].

Recently, biosensors employing nucleic acid receptors have been developed. The underlying mechanism is based on complementary base pairing interaction or on a specific nucleic acid working as an antibody. In particular, aptamers represent the alternative antibody for target recognition. Aptamers are single-stranded (ss) DNA or RNA oligonucleotides that can bind target molecules forming an aptamer/target complex. The link is mediated by molecular complementarity, electrostatic interactions, or hydrogen bonds, characterized by strong affinity and specificity secondary to conformational change [13,59,77]. Aptamers are small-sized molecules that are highly effective as recognition molecules for biosensor systems; this could be attributable to their high chemical stability, high binding affinity and specificity, and simplicity of modification and synthesis [66,67].

Thanks to the aforementioned biochemical features, a highly specific system based on aptamer biosensors with electrochemical recognition have been developed for the identification of LPS [4].

As briefly reported, several commercial techniques are already available for monitoring LPS rapidly and easily, but they are often expensive methods. In addition, in some cases, a combination of several different techniques has been employed to identify, analyze, and measure LPS in biological samples. For example, LPS was determined through reversed-phase HPLC and quantified through MS/MS combined with the LAL test [78]. The combination of these tools gives excellent results in LPS detection and quantification, but it is a complex and expensive method requiring high-technology instruments.

Endotoxin Activity Assay. An excellent technique to measure LPS in a short time (15–20 min) is the endotoxin activity assay (EAA): a rapid test for the detection of endotoxemia in whole blood. EAA is a quick and easy diagnostic test based on a monoclonal antibody that identifies endotoxin. With this method, LPS activity is measured based on the corresponding oxidative burst of primed neutrophils (complexes of an anti-endotoxin antibody and endotoxin) and is detected via the chemiluminescence method [68]. With this approach, reliable quantification of the amount of endotoxin in a patient’s whole blood can be easily obtained.

LPS Detection: Limitation and future options

Heterogeneity in LPS structure gives important limitations on the interpretation of plasma endotoxin assays [79]. Optimal LPS recognition through MD-2–TLR4, the host LPS receptor complex, occurs when the lipid A moiety of LPS has six fatty acyl chains and two phosphates: LPSs produced by some bacteria (i.e., *Pseudomonas aeruginosa)* have five acyl chains and are usually less stimulatory to human cells, other bacteria produce LPSs that are TLR4 agonists, TLR4 antagonists, or nonstimulatory.

Bacteria almost always move from a local site of infection to the bloodstream via lymphatics, reaching circulation through the thoracic duct. Trafficking via lymphatics may allow LPS to bind inhibitory proteins (HDL, chylomicrons) before reaching the blood and complexes may be cleared from the circulation very slowly. LPS molecules bound to those plasma proteins reach the liver, which can remove a significant fraction of the LPS complexes. Therefore, LPS detectable in peripheral venous blood may be a fraction of the total amount.

Inactivating mechanisms operate on mucosal surfaces and in tissues, lymph, and blood, and they may profoundly influence LPS bioactivity, LPS detection in vivo, and the interpretation of plasma LPS assay results. Host enzymes can remove the two decisive signaling structures from lipid A; this process seems to be mediated by phosphatase produced in the small intestine, by macrophages, monocytes, neutrophils, dendritic cells, NK cells, and renal proximal tubule cells. 

Sepsis-induced modulation of neutrophil function might reduce the burst of chemiluminescence used to monitor the response in the EAA. HDL levels typically decrease in the plasma and lymph of septic individuals, but LPS–lipoprotein binding increases nonetheless; in hypertriglyceridemic serum from septic patients, LPS bound mainly to LDL and also to VLDL. Given that there are numerous host mechanisms for enzymatically inactivating lipid A, sequestering lipid A so that it is unable to signal, and inhibiting signaling downstream of TLR4, it seems quite possible that most of the LPS detected in peripheral blood plasma is not stimulatory. 

Given the proven action of inducible nitric oxide synthase inhibitors in targeting mitochondria and reducing oxidative damage in severe sepsis [80], similar to what has been already observed for cardiorenal syndrome type 1 [81], the identification of post-transcriptional modification and amino acid modification could represent an interesting avenue of research for future sepsis biomarkers. Actually, new developments in mass spectrometry offer the opportunity for a more sensitive targeted proteomic approach [80] to identify and quantify ROS-induced modification and the subsequent effects on cellular signaling.

## 5. Sepsis in the Era of Precision Medicine

Even if it is still limited in the area of research on sepsis, an integration between translational bioinformatic resources and targeted treatment would help to stratify patients and improve prognosis [82]. The recent developments in omics technology yield hope for a more detailed understanding of disease pathophysiology. These could further drive *precision medicine* in sepsis [83]. *Precision medicine* aims to match treatment approaches as closely as possible to the patients’ unique individual characteristics, based on biological, genetic, clinical, or other patient data, and to obtain such data as exactly as possible (Figure 3). Opportunities for employing this approach have grown significantly in recent years thanks to the huge amount of data per patient. In this research field belong technologies applied both on the genetic side (genomics, epigenomics, and transcriptomics) and on the molecular side (proteomics and metabolomics). Using omics technology, Sweeney et al. identified and externally validated three subtypes of sepsis: “Inflammopathic”, “Coagulopathic”, and “Adaptive” [84], helping patients’ stratification and, therefore, a more personalized therapeutical approach. The main limitations of the method could be identified on the technical side, such as interference of human DNA, amplification biases, and the need for the effective lysis of all target microbes, and on the clinical side, namely, the high cost and ethical aspects. Translating the results of precision medicine research into routine practice must overcome these barriers, and precision medicine approaches must ensure an equitable impact on the target populations, including ethical considerations, empowering patients to understand this new paradigm in medical practice, and being able to provide or decline informed consent.

## 6. Conclusions

Despite the availability of updated guidelines with the aim of improving outcomes, overall sepsis mortality has increased in the past decade. Understanding underlying mechanisms would help to identify drivers and to prevent reactions to infection. Even if each technology and each approach for endotoxin detection for sepsis diagnosis has its own advantages and limitations, the method used for the identification and quantification of LPS could be selected according to the context and situation. Emerging data and ongoing research on new innovative methods are expected to offer new diagnostic options for LPS evaluation. With the help of these promising technologies, several efforts should be addressed for a more detailed model of sepsis pathways: biomarkers’ identification and their quantification represent the most helpful in a clinical context. Biomarkers allow the development of targeted strategies, as shown by the haemoadsorption of inflammatory molecules, with the aim of early recovery, early recognition of sepsis complications, and improvement in predicting outcomes. In conclusion, considering the wide spectrum of etiology, clinical manifestation, and individual response to infection and therapy, an integrated approach of new technologies and therapies seems to be the most effective strategy to improve outcomes. More research should focus on new biomarkers’ identification and on new technologies for affordable quantification.

## Figures and Tables

**Figure 1 diagnostics-13-00079-f001:**
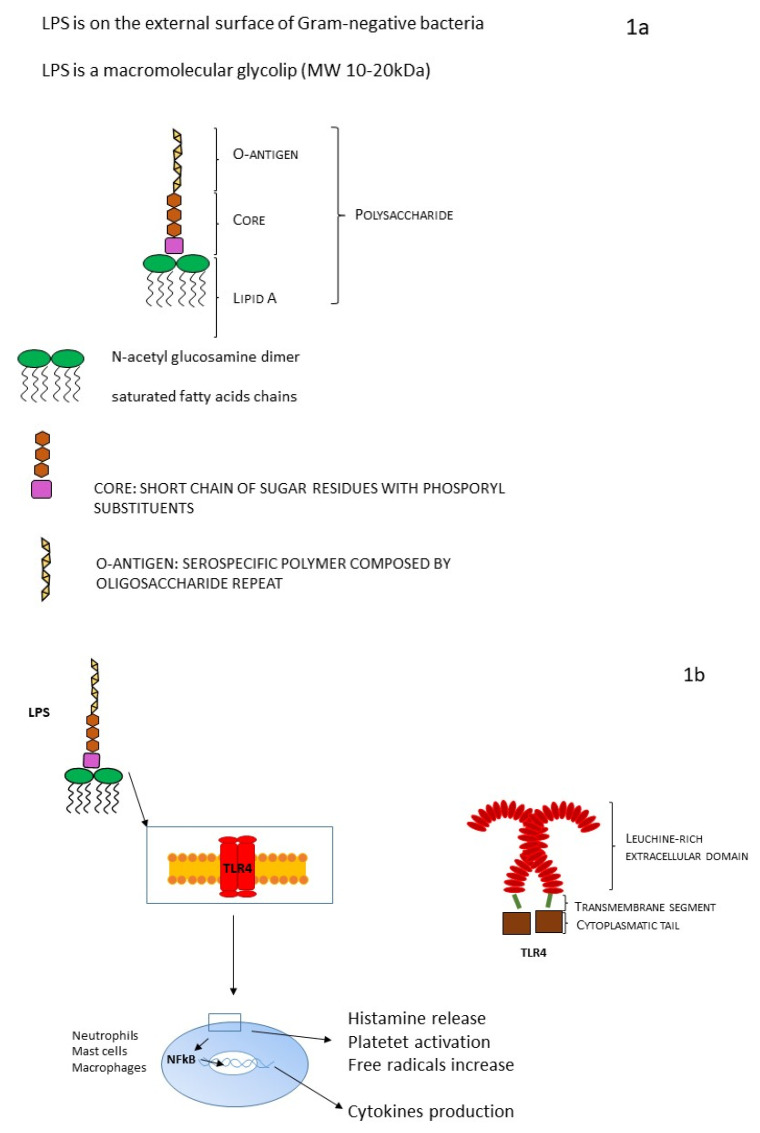
(**a**) LPS structure and (**b**) LPS and TRL mechanism.

**Figure 2 diagnostics-13-00079-f002:**
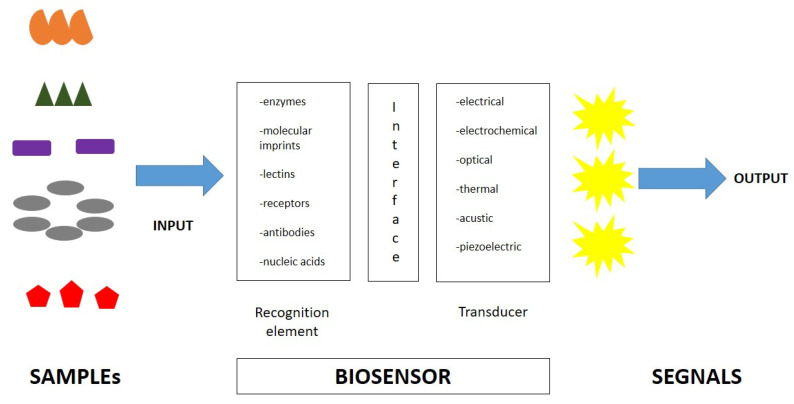
Biosensor characteristics and composition.

**Figure 3 diagnostics-13-00079-f003:**
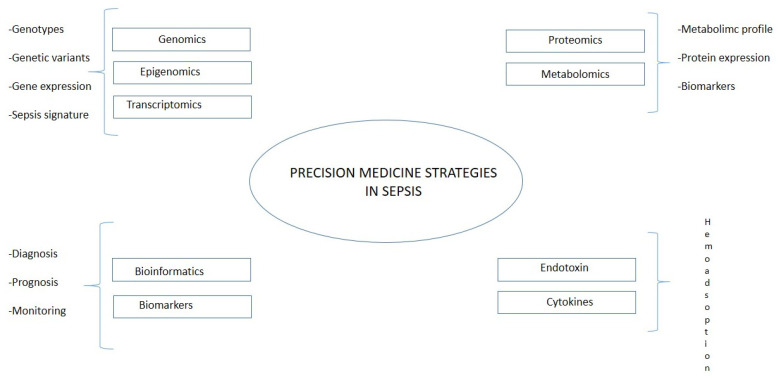
Precision medicine in sepsis.

**Table 1 diagnostics-13-00079-t001:** RCT of LPS extracorporeal removal treatment.

Filter	Study	Clinical Setting	Patients (n)	Endpoints/Outcomes	Status	Year, Place, Reference
Toraymyxin^®^	Multicenter (6) RCT	Severe sepsis after abdominal surgery	17 PMX-B vs. 19 SoC	- No difference in LPS and IL-6 - Increased CI, LVSWI, DO2 - Reduced CRRT need - No difference in SOFA score	Completed	2005, Europe [30]
Toraymyxin^®^	Multicenter (10) RCT [EUPHAS]	Severe sepsis after abdominal surgery	119 PMX-B vs. 113 SoC	- Increased MAP - Reduced inotropic score - PaO(2)/FIO(2) ratio increased - Increased SOFA score - Reduced 28-day mortality	Completed	2004–2007, Italy [31]
Toraymyxin^®^	Multicenter (18) RCT [ABDOMIX]	Severe sepsis after abdominal surgery	34 PMX-B vs. 30 SoC	- No difference 28-day and 90-day mortality - No difference in SOFA score	Completed	2010–2013, France [32]
Toraymyxin^®^	Multicenter (55) RCT [EUPHRATES]	Severe sepsis after abdominal surgery	233 PMX-B vs. 226 SoC	- No difference 28-day and 90-day mortality	Completed	2010–2016, North America [33]
Alteco^®^ LPS adsorber	Multicenter (5) RCT [ASSET]	Severe sepsis of abdominal (20) or urogenital (12) origin	16 LPS Adsorber vs. 16 SoC	Early termination due to patient recruitment issue	Early termination	2015–2016, Northern Europe [34]
oXiris^®^	Monocentric cross over RCT	Septic shock and endotoxin levels > 0.03 EU/mL	10 oXiris vs. 10 SoC	- Reduced LPS - Reduced TNF-α, IL-6, IL-8 and IFNγ - Reduced lactate - Reduced norepinephrine infusion rate	Completed	2016–2018, Belgium [35]
oXiris^®^	Monocentric RCT	Critically ill patients with bleeding risk who underwent anticoagulation-free CRRT	11 oXiris vs. 9 SoC	- Use of oXiris did not prolong filter life over conventional membrane - Significant membrane clogging is observed by 12 h with oXiris	Completed	2012–2016 Singapore [36]
oXiris^®^	Monocentric RCT [ECMORIX]	Cardiogenic shock requiring VA-ECMO	40 oXiris vs. 40 SoC	- Early treatment via the oXiris in cardiogenic shock would allow removal of LPS, thus controlling systemic inflammation, vasoplegia, MOF, and death	Ongoing	2021–2024 (NCT04886180) [37]
oXiris^®^	Monocentric RCT [OXICARD]	Elective cardiac surgery under CPB	35 oXiris vs. 35 SoC	- Primary endpoint: microcirculatory flow index - Secondary endpoints: major cardiovascular and cerebral events, catecholamine use, intensive care unit length of stay, endothelium glycocalyx shedding parameters, cytokines, endothelial biomarkers	Ongoing	France, 2019 (NCT04201119) [38]

RCT: randomized control trial; PMX-B: polymyxin B; LPS: lipopolysaccharide; CI: cardiac index; LVSWI: left ventricular stroke work index; DO2I: oxygen delivery index; CRRT: continuous renal replacement therapy; SOFA: sequential organ failure assessment; MAP: mean artery pressure; TNF-α: tumor necrosis factor-alpha; IFNγ: interferon γ; IL-6: interleukin 6; IL-8: interleukin 8; VA-ECMO: veno-arterial extracorporeal membrane oxygenation; CPB: cardiopulmonary bypass.

**Table 2 diagnostics-13-00079-t002:** LPS tests.

Methods	Principle	Advantages	Limitations	
Rabbit pyrogen test	Increase in rabbit’s temperature after exposition to pyrogenic molecules	First method approved by US Food and Drug Administration	Needs animals Time-consuming Inappropriate for clinical setting Qualitative test	[13]
Limulus amebocyte lysate test (LAL)	Clot formation after exposure of amoebocytes to LPS	User friendly Cheap	Strict experimental conditions Several interfering factors	[14,15,65]
Antibody-based biosensors	Highly specific antigen/antibody affinity (lock and key fit mechanism)	Sensitive Rapid Broad spectrum of target High specificity (if compared to protein-based)	Expensive Time-consuming	[13,57]
Aptames-based biosensors	Base pairing of ss-DNA or RNA forming an aptamer/target complex	Small-sized Highly effective Chemical stability Binding affinity Specificity	Expensive Time-consuming	[66,67]
Endotoxin activity assay (EAA)	Monoclonal antibody against LPS (activity measured through oxidative burst of primed neutrophils)	Short time (15–20 min) Simple Quantitative		[68]

## Data Availability

Not applicable.

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
