# Peer review of "Endotoxin in Sepsis: Methods for LPS Detection and the Use of Omics Techniques"

_diagnostics, 2022, doi:10.3390/diagnostics13010079_

Round 1
Reviewer 1 Report
Comments and suggestions:
1. The article is well written and discussed in details about the endotoxins and its detection.
2. Some areas in the article need modification/improvement/corrections as suggested or indicated in the text. (See the attached manuscript)
3. After moderate revision of this manuscript, it may be fit for publication.

Author Response
Manuscript ID: diagnostics-2026271
Type of manuscript: Review
Title: Endotoxin in sepsis: methods for LPS detection and the use of –omics techniques
Authors: Grazia Maria Virzì *, Maria Mattiotti, Massimo De cal, Claudio Ronco, Monica Zanella, Silvia De Rosa
We thank the reviewer1 for his/her insightful comments. We corrected all points in the text and we responded to all comments in pdf files. We thank the reviewer for the opportunity to clarify these points.We believe the manuscript has been improved with the suggested changes. We hope that now our revised manuscript is acceptable for publication. If there is any further information required, please do not hesitate to contact us.
Sincerely,

Reviewer 2 Report
The review written by Virzì and colleagues has been designed to offer an approach to the study of endotoxemia in sepsis. In general, although the assays used to detect endotoxemia have restrictions and are controversial to consider whether the endotoxin found is bioactive in vivo, assessing the levels of endotoxin in plasma, serum or tissues of individuals are relevant tools for the clinical evolution of septic shock.
Major concerns:
i. The heterogeneity in the structure, solubility, physical state and bioactivity of LPS provides important restrictions in the interpretation of plasma endotoxin assays. The article did not consider how optimal recognition of LPS might be associated with human cells, where the variability in LPS structure may be TLR4 agonists, TLR4 antagonists or non-stimulators. The article will benefit from further explanation relevant to the limitations, considering that lipid A structures can be detected by animal cells, but are not necessarily distinguished by endotoxin assays. Additionally, it is relevant to think that the endotoxin that reaches the portal venous circulation passes through the microcirculation of the organs before reaching a peripheral vein, which is the basis for most endotoxin assays. Authors need to consider that when LPS is detectable in peripheral venous blood, this may be the fraction because it was linked to lipoproteins or altered by clearance. Therefore, it may not be a representative sample of the LPS that initially entered the blood stream or was removed by the tissues.
ii. The description of the structural components of endotoxin mentioned on page 3 (lines 124-175) is not clear to the reader. I suggest adding a figure. Also, the references could be more specific. The authors used one or two references at the end of each paragraph for different contents; it is worth mentioning that the text was not built in consensus with different studies; i.e., a single source (“review article”) rather than the main base (“research article”).
iii Studies seem to show that LPS levels decrease in parallel with therapy when it is effective. However, it is uncertain whether clinical improvement can be enhanced by treatments designed to antagonize endotoxins as discussed in item 3 “Removal of endotoxins by extracorporeal therapies”. The addition of a table with endotoxin elimination treatment with successful and unsuccessful results from previous studies could serve to understand projects involving groups of septic patients.
iv. In the overall scenario of research efforts dedicated to analysis of endotoxemia in sepsis, the study of reactive oxygen species (ROS) may be responsible for tissue injury. This is an important aspect that must be, at the very least, pointed out. Redox receptors have a structure that can essentially contribute to rapid mechanisms designed to deal with ROS and make critical adjustments that allow bacteria to survive. Some biosensor systems currently available for detecting such aberrant oxidative stress are confined to probing indirect surrogate markers such as ROS-induced by-products.
v. Table 1 may contain in the fourth column the title “References” and below only the number corresponding to the cited studies. Please check the number of references. *Tavener, Angus and Steinhagen, and successively others, are not reference 11,12,13…. However, in my view, the table as shown it is almost redundant in a concise way considering the text (line 293 to 388). A table should contain important informations that might otherwise unnecessarily lengthen a paper and, hence, is usefully referenced from the main text. If the idea is a synthesis, perhaps this could be added as supplemental material.
Minor comments:
Ø Page 3 – (Line 98-106) is without references;
Ø Page 5 – (Line 223) Authors cited references to techniques: genomics, transcriptomics, proteomics, but not to metabolomics;
Ø Figure 2 "proposed research scheme for the advancement of precision medicine in sepsis" is not well located in the text. It would be interesting to change to line 396 at the end of the sentence “Precision medicine has the aim to… exactly as possible (Figure 2).
Author Response
Manuscript ID: diagnostics-2026271
Type of manuscript: Review
Title: Endotoxin in sepsis: methods for LPS detection and the use of –omics techniques
Authors: Grazia Maria Virzì *, Maria Mattiotti, Massimo de Cal, Claudio Ronco, Monica Zanella, Silvia De Rosa
Comments and Suggestions for Authors
We thank the reviewer2 for his/her insightful comments.
The review written by Virzì and colleagues has been designed to offer an approach to the study of endotoxemia in sepsis. In general, although the assays used to detect endotoxemia have restrictions and are controversial to consider whether the endotoxin found is bioactive in vivo, assessing the levels of endotoxin in plasma, serum or tissues of individuals are relevant tools for the clinical evolution of septic shock.
We thank the reviewer for these observations.
Major concerns:
- The heterogeneity in the structure, solubility, physical state and bioactivity of LPS provides important restrictions in the interpretation of plasma endotoxin assays. The article did not consider how optimal recognition of LPS might be associated with human cells, where the variability in LPS structure may be TLR4 agonists, TLR4 antagonists or non-stimulators. The article will benefit from further explanation relevant to the limitations, considering that lipid A structures can be detected by animal cells, but are not necessarily distinguished by endotoxin assays. Additionally, it is relevant to think that the endotoxin that reaches the portal venous circulation passes through the microcirculation of the organs before reaching a peripheral vein, which is the basis for most endotoxin assays. Authors need to consider that when LPS is detectable in peripheral venous blood, this may be the fraction because it was linked to lipoproteins or altered by clearance. Therefore, it may not be a representative sample of the LPS that initially entered the blood stream or was removed by the tissues.
We thank reviewer for this punctual observation. We did not highlight main limitations due to the intrinsic nature of LPS and its possible false results obtained with available methods. We add a brief paragraph with a more detailed explanation of limitations.
- The description of the structural components of endotoxin mentioned on page 3 (lines 124-175) is not clear to the reader. I suggest adding a figure. Also, the references could be more specific. The authors used one or two references at the end of each paragraph for different contents; it is worth mentioning that the text was not built in consensus with different studies; i.e., a single source (“review article”) rather than the main base (“research article”).
We thank the reviewer for the suggestions. We further explain and clarify with a more detailed description LPS structure. In order to make explanation more feasible, we add, as suggested, a figure that simplify macroscopic structure of LPS. We furthermore extend bibliography as suggested. Our previous choice to refer to review was based on old bibliography available about LPS chemical structure.
- Studies seem to show that LPS levels decrease in parallel with therapy when it is effective. However, it is uncertain whether clinical improvement can be enhanced by treatments designed to antagonize endotoxins as discussed in item 3 “Removal of endotoxins by extracorporeal therapies”. The addition of a table with endotoxin elimination treatment with successful and unsuccessful results from previous studies could serve to understand projects involving groups of septic patients.
As suggested by the reviewer we add a Table (Table1) with main RCTs about LPS removal in different clinicals settings. We hope now the work and the reference could be more complete and clear
- In the overall scenario of research efforts dedicated to analysis of endotoxemia in sepsis, the study of reactive oxygen species (ROS) may be responsible for tissue injury. This is an important aspect that must be, at the very least, pointed out. Redox receptors have a structure that can essentially contribute to rapid mechanisms designed to deal with ROS and make critical adjustments that allow bacteria to survive. Some biosensor systems currently available for detecting such aberrant oxidative stress are confined to probing indirect surrogate markers such as ROS-induced by-products.
We thank the reviewer for the interesting topic. As suggested we add a link to the role of mass spectrometry in ROS induced modification, and his possible promising application in sepsis condition.
- Table 1 may contain in the fourth column the title “References” and below only the number corresponding to the cited studies. Please check the number of references. *Tavener, Angus and Steinhagen, and successively others, are not reference 11,12,13…. However, in my view, the table as shown it is almost redundant in a concise way considering the text (line 293 to 388). A table should contain important informations that might otherwise unnecessarily lengthen a paper and, hence, is usefully referenced from the main text. If the idea is a synthesis, perhaps this could be added as supplemental material.
We thank the reviewer for this observation. We have now modified and corrected and updated references of Table 1. Our main aim was to give to the reader a synthetic view of the main available technics to have a more practical material for a fast consultation.
Minor comments:
- Page 3 – (Line 98-106) is without references;
We thank the reviewer for this observation. We have now added references.
- Page 5 – (Line 223) Authors cited references to techniques: genomics, transcriptomics, proteomics, but not to metabolomics;
We thank the reviewer for this observation. We have now added references.
- Figure 2 "proposed research scheme for the advancement of precision medicine in sepsis" is not well located in the text. It would be interesting to change to line 396 at the end of the sentence “Precision medicine has the aim to… exactly as possible (Figure 2).
We thank the reviewer for this suggestion. We have now relocated Figure 2..
We thank the reviewer1 for his/her insightful comments. We corrected all points in the text and we responded to all comments in pdf files. We thank the reviewer for the opportunity to clarify these points.We believe the manuscript has been improved with the suggested changes. We hope that now our revised manuscript is acceptable for publication. If there is any further information required, please do not hesitate to contact us.
Sincerely,
